# CEDNet: A Cascade Encoder-Decoder Network for Dense Prediction

## ABSTRACT

Multi-scale features are essential for dense prediction tasks, such as object detection, instance segmentation, and semantic segmentation. The prevailing methods usually utilize a classification backbone to extract multi-scale features and then fuse these features using a lightweight module (*e.g.*, the fusion module in FPN and BiFPN, two typical object detection methods). However, as these methods allocate most computational resources to the classification backbone, the multi-scale feature fusion in these methods is delayed, which may lead to inadequate feature fusion. While some methods perform feature fusion from early stages, they either fail to fully leverage high-level features to guide low-level feature learning or have complex structures, resulting in sub-optimal performance. We propose a streamlined cascade encoder-decoder network, dubbed CEDNet, tailored for dense prediction tasks. All stages in CEDNet share the same encoder-decoder structure and perform multi-scale feature fusion within the decoder. A hallmark of CEDNet is its ability to incorporate high-level features from early stages to guide low-level feature learning in subsequent stages, thereby enhancing the effectiveness of multi-scale feature fusion. We explored three well-known encoder-decoder structures: Hourglass, UNet, and FPN. When integrated into CEDNet, they performed much better than traditional methods that use a pre-designed classification backbone combined with a lightweight fusion module. Extensive experiments on object detection, instance segmentation, and semantic segmentation demonstrated the effectiveness of our method. The code will be made publicly available.

## 1 INTRODUCTION

In recent years, both convolutional neural networks (CNNs) and transformer-based networks have achieved remarkable results in various computer vision tasks, including image classification, object detection, and semantic segmentation. In image classification, the widely-used CNNs (Krizhevsky et al., 2012; Simonyan & Zisserman, 2015; Szegedy et al., 2015; He et al., 2016; Liu et al., 2022) as well as the recently developed transformer-based networks (Liu et al., 2021; Yang et al., 2021; Dong et al., 2022; Zhang et al., 2023b) generally follow a sequential architectural design. They progressively reduce the spatial size of feature maps and make predictions based on the coarsest scale of features. However, in dense prediction tasks, such as object detection and instance segmentation, the need for multi-scale features arises to accommodate objects of diverse sizes. Therefore, effectively extracting and fusing multi-scale features becomes essential for the success of these tasks (He et al., 2017; Lin et al., 2017b; Tian et al., 2019; Xiao et al., 2018; Zhang et al., 2021; Hu et al., 2022).

Many methods have been proposed for multi-scale feature extraction and fusion (Lin et al., 2017a; Liu et al., 2018; Ghiasi et al., 2019; Tan et al., 2020). One widely-used model is the feature pyramid network (FPN) (Lin et al., 2017a) (Figure 1 (a)). FPN consists of a pre-designed classification backbone for extracting multi-scale features and a lightweight fusion module for fusing these features. Moving beyond the FPN, some cascade fusion strategies have been developed and showcased efficacy in multi-scale feature fusion (Liu et al., 2018; Ghiasi et al., 2019; Tan et al., 2020). Figure 1 (b) shows the structure of the representative BiFPN (Tan et al., 2020). It iteratively fuses multi-scale features using repeated bottom-up and top-down pathways. However, the time for feature fusion in these networks is relatively late, because they allocate most computational resources to the classification backbone to extract the initial multi-scale features. We define *the time for feature fusion* as the ratio of the parameters of the sub-network before the first fusion module to the whole network. A

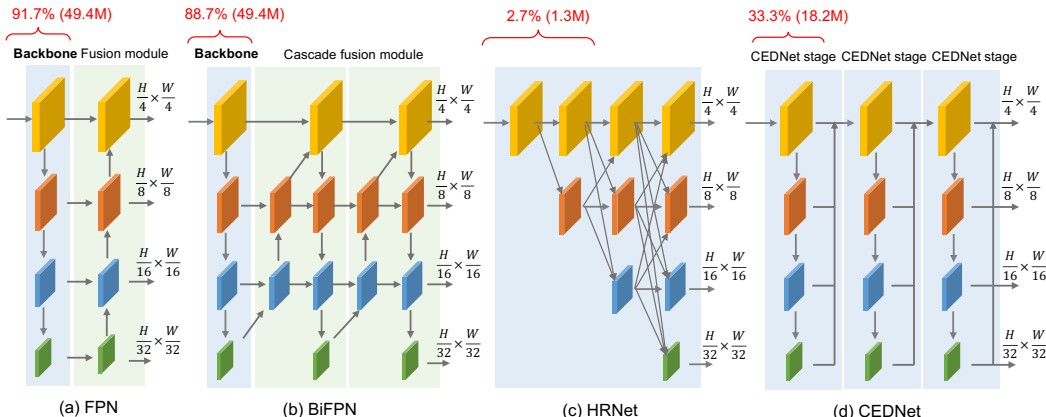

Figure 1: Comparison among FPN, BiFPN, HRNet, and our CEDNet. $H \times W$ denotes the spatial size of the input image. At the top of each panel, the percentage indicates the time to perform the first multi-scale feature fusion, while the number in bracket is the number of parameters of the selected part. For these calculations, we take ConvNeXt-S as the backbone of FPN and BiFPN. While we illustrate a CEDNet of four scales here for a clearer comparison, it's noteworthy that in our actual implementation, feature maps with a resolution of $\frac{H}{4} \times \frac{W}{4}$ are not included in the CEDNet stages.

smaller ratio indicates an earlier time. For instance, considering an FPN built based on the classification model ConvNeXt-S (Liu et al., 2022), the time for feature fusion is the ratio of the parameters of the backbone ConvNeXt-S to the entire FPN (91.7%). Given the complexity of dense prediction tasks where models are required to handle objects of diverse sizes, we expect that integrating early multi-scale feature fusion within the backbone could enhance model performance.

Some methods have transitioned from using pre-designed classification networks to designing task-specific backbones for dense prediction tasks (Wang et al., 2019; Du et al., 2020; Jiang et al., 2022; Cai et al., 2023). In these methods, some incorporate early multi-scale feature fusion. For example, HRNet (Wang et al., 2019), one of the representative works (Figure 1 (c)), aims to learn semantically rich and spatially precise features. Although HRNet performs the first feature fusion very early (2.7%), it generates high-level (low-resolution) features with strong semantic information quite late. This limits their role in guiding the learning of low-level (high-resolution) features that are important for dense prediction tasks. In contrast, SpineNet (Du et al., 2020) employs neural architecture search (NAS) (Zoph & Le, 2017) to learn a scale-permuted backbone with early feature fusion. Nevertheless, the resulting network is complex and exhibits limited performance when transferred to different detectors (Du et al., 2020). GiraffeDet (Jiang et al., 2022) integrates a lightweight backbone with a heavy fusion module for object detection, aiming to enhance the information exchange between high-level and low-level features. Yet, it fuses multi-scale features in a fully connected way, which inevitably increases runtime latency. A detailed discussion of related works can be found in Section 2.2. Clearly, an appropriate structure for effective early multi-scale feature fusion is lacking.

In this paper, we present CEDNet, a cascade encoder-decoder network tailored for dense prediction tasks. CEDNet begins with a stem module to extract initial high-resolution features. Following this, CEDNet incorporates several cascade stages to generate multi-scale features, with all stages sharing the same encoder-decoder structure. The encoder-decoder structure can be realized in various ways. Figure 1 (d) illustrates a three-stage CEDNet built on the FPN-style design. CEDNet evenly allocates its computational resources across stages and fuses multi-scale features within each decoder. As a result, CEDNet performs multi-scale feature fusion from the early stages of the network. This strategy ensures that high-level features from the early stages are integrated to guide the learning of low-level features in subsequent stages. Moreover, CEDNet possesses a more streamlined and efficient structure, making it suitable for a wide variety of models and tasks.

We investigated three well-known methods, *i.e.*, Hourglass (Newell et al., 2016), UNet (Ronneberger et al., 2015), and FPN (Ghiasi et al., 2019), as the encoder-decoder structure in experiments and found that they all performed well. Due to the slightly better results of the FPN, it is adopted as the default encoder-decoder structure in CEDNet for further analysis on object detection, instance seg-

mentation and semantic segmentation. On the COCO *val 2017* for object detection and instance segmentation, the CEDNet variants outperformed their counterparts, *i.e.*, the ConvNeXt variants (Liu et al., 2022), achieving an increase of 1.9-2.9 % in box AP and 1.2-1.8% in mask AP based on the popular framework RetinaNet (He et al., 2016) and Mask R-CNN (He et al., 2017). On the ADE20k for semantic segmentation, the CEDNet variants outperformed their counterparts by 0.8-2.2% mIoU based on the renowned framework UperNet (Xiao et al., 2018). These results demonstrate the excellent performance of CEDNet and encourage the community to rethink the prevalent model design principle for dense prediction tasks.

## 2 RELATED WORK

### 2.1 MULTI-SCALE FEATURE FUSION

Many methods adopt pre-designed classification backbones to extract multi-scale features. However, the low-level features produced by traditional classification networks are semantically weak and ill-suited for downstream dense prediction tasks. To tackle this limitation, many strategies (Lin et al., 2017a; Liu et al., 2018; Tan et al., 2020; Chen et al., 2018; Ghiasi et al., 2019) have been proposed for multi-scale feature fusion. In semantic segmentation, DeeplabV3+ (Chen et al., 2018) fuses low-level features with semantically strong high-level features produced by atrous spatial pyramid pooling. In object detection, FPN (Lin et al., 2017a) introduces a top-down pathway to sequentially combine high-level features with low-level features. NAS-FPN (Ghiasi et al., 2019) fuses multi-scale features by repeated fusion stages searched by neural architecture search (Zoph & Le, 2017). EfficientDet (Tan et al., 2020) adopts a weighted bi-directional feature pyramid network in conjunction with a compound scaling rule to achieve efficient feature fusion. A common drawback of these methods is that they allocate most computational resources to the classification backbone, delaying feature fusion and potentially undermining fusion effectiveness. In contrast, our approach evenly allocates computational resources to multiple stages and perform feature fusion within each stage.

### 2.2 BACKBONE DESIGNS FOR DENSE PREDICTION

Instead of fusing multi-scale features from pre-designed classification networks, some studies have attempted to design task-specific backbones for dense prediction tasks (Ronneberger et al., 2015; Newell et al., 2016; Wang et al., 2019; Du et al., 2020; Jiang et al., 2022; Liu et al., 2020; Qiao et al., 2021; Cai et al., 2023). For instance, UNet (Ronneberger et al., 2015) employs a U-shape structure to acquire high-resolution and semantically strong features in medical image segmentation. Hourglass (Newell et al., 2016) introduces a convolutional network consisting of repeated bottom-up and top-down pathways for human pose estimation. HRNet (Wang et al., 2019) retains high-resolution features throughout the whole network and performs well in semantic segmentation and human pose estimation. In object detection, SpineNet (Du et al., 2020) leverages neural architecture search (Zoph & Le, 2017) to learn scale-permuted backbones. GiraffeDet (Jiang et al., 2022) pairs a lightweight backbone with a heavy fusion module to encourage dense information exchange among multi-scale features. RevCol (Cai et al., 2023) feeds the input image to several identical subnetworks simultaneously and connects them through reversible transformations.

While the aforementioned methods incorporate early multi-scale feature fusion, they either exhibit effectiveness solely on specific models and tasks (Ronneberger et al., 2015; Newell et al., 2016; Du et al., 2020), or do not fully harness the potential of high-level features to guide the learning of low-level features (Wang et al., 2019; Cai et al., 2023). In contrast, our method incorporates multiple cascade stages to iteratively extract and fuse multi-scale features. Therefore, the high-level features from early stages can be integrated to instruct the learning of low-level features in subsequent stages. Moreover, CEDNet showcases excellent performance across a broad spectrum of models and tasks.

## 3 CEDNET

### 3.1 OVERALL ARCHITECTURE

Figure 2 illustrates the overall architecture of CEDNet. The input RGB image with a spatial size of $H \times W$ is fed into a stem module to extract high-resolution feature maps of size $\frac{H}{8} \times \frac{W}{8}$. The

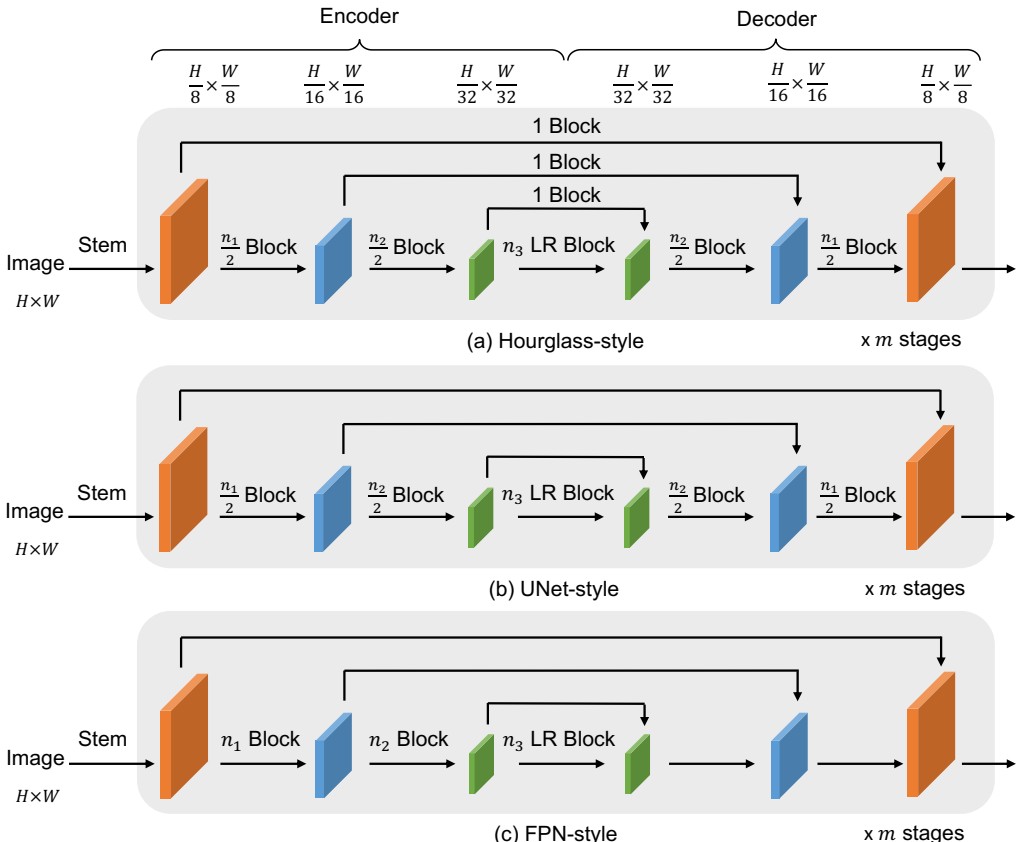

Figure 2: Three implementations of CEDNet. The input image with a spatial size of $H \times W$ is fed into a lightweight stem module to extract high-resolution features of size $\frac{H}{8} \times \frac{W}{8}$. These features are then processed through $m$ cascade stages to extract multi-scale features. Block denotes CED block, and LR Block denotes long-range (LR) CED block. The down-sampling layers ($2 \times 2$ convolution with stride 2) and the up-sampling layers (bilinear interpolation) are omitted for clarity.

stem module comprises two sequential $3 \times 3$ convolutional layers (each with a stride of 2), $n_0$ CED blocks, and a $2 \times 2$ convolutional layer with a stride of 2. Each $3 \times 3$ convolutional layer is followed by a LayerNorm (Ba et al., 2016) layer and a GELU (Hendrycks & Gimpel, 2016) unit. The further details about the CED block can be found in Section 3.3. Subsequently, $m$ cascade stages, each with the same encoder-decoder structure, are utilized to extract multi-scale features. The multi-scale features from the final decoder are then fed into downstream dense prediction tasks. *Unlike in FPN and BiFPN, no extra feature fusion modules are required after the CEDNet backbone.* We discuss three implementations of the encoder-decoder structure in Section 3.2.

## 3.2 THREE ENCODER-DECODER STRUCTURES

In CEDNet, each stage employs an encoder-decoder structure. The encoder extracts multi-scale features, while the decoder integrates these features into single-scale, highest-resolution ones. Consequently, the high-level (low-resolution) features from early stages are integrated to guide the learning of low-level features in subsequent stages. While many methods can be used to realize the encoder-decoder structure, we adopt three well-known methods for our purposes in this study.

**Hourglass-style.** The Hourglass network (Newell et al., 2016) is a deep learning architecture specifically designed for human pose estimation. It resembles an encoder-decoder design but stands out with its symmetrical hourglass shape, from which its name is derived. In this study, we draw inspiration from the Hourglass architecture to devise an hourglass-style encoder-decoder (Figure 2 (a)). In alignment with the original design, a CED block is employed to transform the feature maps from the encoder before integrating them into the symmetrical feature maps in the decoder.

**UNet-style.** UNet is a prominent network primarily employed in medical image segmentation (Ronneberger et al., 2015). Recent advancements have also shown its successful application in diffusion models (Zhang et al., 2023a). As illustrated in Figure 2 (b), the UNet-style encoder-decoder has a symmetrical shape. Unlike the hourglass-style design, identity skip connections are harnessed to bridge the symmetrical feature maps between the encoder and the decoder.

**FPN-style.** FPN (Lin et al., 2017a) is initially designed for object detection and instance segmentation, aiming to fuse multi-scale features from pre-designed classification networks. In this work, we incorporate the FPN-style encoder-decoder as a separate stage in CEDNet, as shown in Figure 2 (c). Different from the standard FPN implementation, we eliminate the $3 \times 3$ convolutions responsible for transforming the merged symmetrical feature maps. As a result, most computational resources are allocated to the encoders, with only two $1 \times 1$ convolutions in each decoder for feature channel alignment.

### 3.3 BLOCK DESIGNS

**CED block.** *The solid elements* in Figure 3 illustrate the general structure of the CED block. This block comprises a token mixer for spatial feature interactions and a multi-layer perceptron (MLP) with two layers for channel feature interactions. The token mixer can be various existing designs, such as the $3 \times 3$ convolution in ResNet (He et al., 2016), the $7 \times 7$ depth-wise convolution in ConvNeXt (Liu et al., 2022), and the local window attention in Swin transformer (Liu et al., 2021). In CEDNet, we take the lightweight $7 \times 7$ depth-wise convolution from ConvNeXt as the default token mixer. *Please note that a more powerful token mixer may yield enhanced performance, but that is not the focus of this work.*

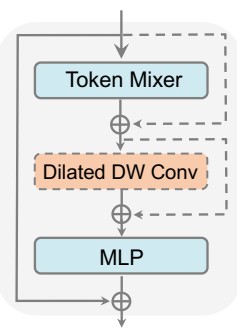

**LR CED block.** To increase the receptive field of neurons, we introduce the LR CED block. Beyond the CED block, this block incorporates a $7 \times 7$ *dilated* depth-wise convolution accompanied by two skip connections, as highlighted by the dashed elements. By integrating the dilated depth-wise convolution, the LR CED block is capable of capturing long-range dependencies among spatial features with only a marginal increase in parameters and computational overhead. The LR CED blocks are utilized to transform the lowest-resolution features in each CEDNet stage.

Figure 3: Structure of the CED block and the LR CED block. The CED block consists solely of solid elements, while the LR CED block includes both solid and dashed elements. DW Conv is short for depth-wise convolution.

### 3.4 ARCHITECTURE SPECIFIACTIONS

We have constructed three CEDNet variants based on the FPN-style encoder-decoder, *i.e.*, CEDNet-NeXt-T/S/B, where the suffixes T/S/B indicate the model size tiny/small/base. We take the $7 \times 7$ depth-wise convolution from ConvNeXt as the default token mixer for all (LR) CED blocks. For all LR CED blocks, we set the dilation rate $r$ of the dilated convolution to 3. These variants adopt different channel dimensions $C$, different numbers of blocks $B = (n_0, n_1, n_2, n_3)$, and different numbers of stages $m$. The configuration hyper-parameters for these variants are presented below:

- CEDNet-NeXt-T: $C$=(96, 192, 352, 512), $B$=(3, 2, 4, 2), $m$=3
- CEDNet-NeXt-S: $C$=(96, 192, 352, 512), $B$=(3, 2, 7, 2), $m$=4
- CEDNet-NeXt-B: $C$=(128, 256, 448, 704), $B$=(3, 2, 7, 2), $m$=4

## 4 EXPERIMENTS

### 4.1 THREE ENCODER-DECODER STRUCTURES

We conducted experiments to compare the three encoder-decoder structures.

**Pre-training settings.** Following common practice (Liu et al., 2021; 2022), we pre-trained CEDNet on the ImageNet-1K dataset (Deng et al., 2009). The ImageNet-1K dataset consists of 1000 object classes with 1.2M training images. To perform classification, we removed the last decoder

Table 1: Comparison among three encoder-decoder structures. $AP^b$ is the overall detection accuracy on COCO *val2017*. The model inference speed FPS was measured on a single RTX 3090 GPU.

| Method | Param | FLOPs | $AP^b$ | $AP^b_{50}$ | $AP^b_{75}$ | FPS↑ |
|---|---|---|---|---|---|---|
| ConvNeXt-T w/ FPN (Lin et al., 2017a) | 39M | 243G | 45.4 | 66.5 | 48.7 | 20.0 |
| ConvNeXt-T w/ NAS-FPN (Ghiasi et al., 2019) | 47M | 289G | 46.6 | 66.8 | 49.9 | 18.1 |
| ConvNeXt-T w/ BiFPN Tan et al. (2020) | 39M | 248G | 46.7 | 67.1 | 50.2 | 16.8 |
| HRNet-w32 (Wang et al., 2019) | 39M | 320G | 45.7 | 66.5 | 49.2 | 15.9 |
| SpineNet-96 (Du et al., 2020) | 43M | 265G | 47.1 | 67.1 | 51.1 | 16.3 |
| GiraffeDet-D11 (Jiang et al., 2022) | 69M | 275G | 46.6 | 65.0 | 51.1 | 12.4 |
| CEDNet-NeXt-T (Hourglass-style) | 39M | 255M | 47.4 | 68.5 | 50.7 | 15.9 |
| CEDNet-NeXt-T (UNet-style) | 39M | 255M | 47.9 | 68.9 | 51.4 | 16.7 |
| CEDNet-NeXt-T (FPN-style) | 39M | 255M | **48.3** | **69.1** | **51.6** | 17.1 |

and attached a classification head on the lowest-resolution features from the last stage. The hyper-parameters, augmentation and regularization strategies strictly follows (Liu et al., 2022).

**Pre-training results.** We built the CEDNet models using three different encoder-decoder structures. Appendix A presents the results of the CEDNet models on ImageNet-1K in comparison with some recent methods in image classification. We report the top-1 accuracy on the validation set. Table A1 shows that the CEDNet models slightly outperformed their counterparts, *i.e.*, the ConvNeXt variants. *Please note that the CEDNet is specifically designed for dense prediction tasks, and surpassing state-of-the-art methods in image classification is not our goal.*

**Fine-tuning settings.** We fine-tuned models on object detection with COCO 2017 (Lin et al., 2014) based on the well-known detection framework RetinaNet (Lin et al., 2017b) using the MMDetection toolboxes (Chen et al., 2019). For training settings, we mainly followed (Liu et al., 2022). Additionally, we found that the proposed CEDNet models were easy to overfit the training data. To fully explore the potential of CEDNet models, we used large scale jittering and copy-and-paste data augmentation following (Ghiasi et al., 2021), but only with box annotations. We re-trained all baseline models with the same data augmentation for a fair comparison.

**Fine-tuning results.** Table 1 shows that all three CEDNet models yielded significant gains over the models with FPN, NAS-FPN, and BiFPN, all of which utilize the classification network ConvNeXt-T to extract initial multi-scale features. This result validates the effectiveness of the cascade encoder-decoder network that performs multi-scale feature fusion from early stages. In addition, the CED-Net models surpassed other early feature fusion methods: HRNet, SpineNet, and GiraffeDet. We attempted to pre-train the entire BiFPN model by attaching a classification head to the coarsest feature maps of the last bottom-up pathway in the fusion module. However, we obtained poor results, *i.e.*, 76.8% top-1 accuracy on ImageNet and 39.4% box AP on COCO.

Since the model built on the FPN-style encoder-decoder slightly outperformed the models built on the UNet-style and Hourglass-style encoder-decoder in both detection accuracy and model inference speed, *we adopted the FPN-style encoder-decoder for CEDNet by default in subsequent experiments.*

## 4.2 OBJECT DETECTION ON COCO

**Settings.** We benchmark our models on object detection with COCO 2017 (Lin et al., 2014) based on four representative frameworks, *i.e.*, Deformable DETR (Zhu et al., 2021), RetinaNet (Lin et al., 2017b), Mask R-CNN (He et al., 2017), and Cascade Mask R-CNN (Cai & Vasconcelos, 2018). All training settings were same as the fine-tuning settings in Section 4.1.

**Main results.** Table 2 presents the object detection results of the CEDNet models to compare with other methods. The CEDNet models yielded significant gains over the ConvNeXt models. Specifically, CEDNet-NeXt-T achieved 2.2%, 2.9%, 2.8%, and 1.7% box AP improvements over its counterpart ConvNeXt-T based on the Deformable DETR, RetinaNet, Mask R-CNN, and Cascade Mask R-CNN, respectively. When scaled up to CEDNet-NeXt-S, CEDNet still outperformed its baseline ConvNeXt-S by 1.3%, 2.2%, 1.9%, and 1.6% box AP based on the four detectors.

Table 2: Results of object detection and instance segmentation on the COCO *val2017*. $AP^b$ and $AP^m$ are the overall metrics for object detection and instance segmentation, respectively. If required, FPN was adopted as the default fusion module for methods except CEDNet and the models marked by $\dagger$

(a) Deformable DETR

| Method | Param | FLOPs | $AP^b$ | $AP^b_{50}$ | $AP^b_{75}$ | $AP^b_S$ | $AP^b_M$ | $AP^b_L$ |
|---|---|---|---|---|---|---|---|---|
| ConvNeXt-T | 42M | 231G | 47.1 | 66.7 | 51.8 | 28.8 | 50.5 | 62.3 |
| ConvNeXt-S | 64M | 317G | 49.0 | 68.6 | 53.7 | 31.2 | 52.4 | 64.5 |
| CEDNet-NeXt-T | 43M | 223G | 49.3 | 69.1 | 53.7 | 32.1 | 52.8 | 65.3 |
| CEDNet-NeXt-S | 65M | 304G | 50.3 | 70.2 | 55.2 | 32.3 | 54.6 | 65.2 |

(b) RetinaNet

| Backbone | Param | FLOPs | $AP^b$ | $AP^b_{50}$ | $AP^b_{75}$ | $AP^b_S$ | $AP^b_M$ | $AP^b_L$ |
|---|---|---|---|---|---|---|---|---|
| SpineNet-143$^\dagger$ (Du et al., 2020) | 67M | 524G | 48.1 | 67.6 | 52.0 | 30.2 | 51.1 | 59.9 |
| Swin-T (Liu et al., 2021) | 39M | 245G | 45.0 | 65.9 | 48.4 | 29.7 | 48.9 | 58.1 |
| Swin-S (Liu et al., 2021) | 60M | 335G | 46.4 | 67.0 | 50.1 | 31.0 | 50.1 | 60.3 |
| Swin-B (Liu et al., 2021) | 98M | 477G | 45.8 | 66.4 | 49.1 | 29.9 | 49.4 | 60.3 |
| ConvNeXt-T | 39M | 243G | 45.4 | 67.0 | 48.7 | 29.5 | 49.9 | 59.9 |
| ConvNeXt-S | 60M | 329G | 47.4 | 68.3 | 51.2 | 32.0 | 51.5 | 61.6 |
| CEDNet-NeXt-T | 39M | 255G | 48.3 | 69.1 | 51.6 | 33.2 | 53.1 | 62.7 |
| CEDNet-NeXt-S | 61M | 335G | 49.6 | 70.8 | 53.2 | 34.8 | 54.0 | 63.5 |

(c) Mask R-CNN

| Backbone | Param | FLOPs | $AP^b$ | $AP^b_{50}$ | $AP^b_{75}$ | $AP^m$ | $AP^m_{50}$ | $AP^m_{75}$ |
|---|---|---|---|---|---|---|---|---|
| DetectoRS-50$^\dagger$ (Qiao et al., 2021) | 105M | 432G | 46.2 | 65.1 | 50.2 | 40.4 | 62.5 | 43.5 |
| Swin-T (Liu et al., 2021) | 48M | 264G | 46.0 | 68.1 | 50.3 | 41.6 | 65.1 | 44.9 |
| Swin-S (Liu et al., 2021) | 69M | 354G | 48.5 | 70.2 | 53.5 | 43.3 | 67.3 | 46.6 |
| FocalNet-S (Yang et al., 2022) | 72M | 365G | 49.3 | 50.9 | 54.6 | 44.1 | 67.9 | 47.4 |
| Swin-B (Liu et al., 2021) | 107M | 496G | 48.5 | 69.8 | 53.2 | 43.4 | 66.8 | 46.9 |
| FocalNet-B (Yang et al., 2022) | 114M | 507G | 49.8 | 70.7 | 54.2 | 43.8 | 68.2 | 47.2 |
| ConvNeXt-T | 48M | 262G | 46.4 | 68.1 | 51.3 | 42.3 | 65.2 | 45.9 |
| ConvNeXt-S | 70M | 348G | 48.5 | 70.0 | 53.3 | 43.8 | 67.2 | 47.7 |
| CEDNet-NeXt-T | 49M | 274G | 49.2 | 70.3 | 53.7 | 44.1 | 67.8 | 47.5 |
| CEDNet-NeXt-S | 72M | 355G | 50.4 | 71.7 | 55.1 | 45.0 | 68.9 | 48.6 |

(d) Cascade Mask R-CNN

| Backbone | Param | FLOPs | $AP^b$ | $AP^b_{50}$ | $AP^b_{75}$ | $AP^m$ | $AP^m_{50}$ | $AP^m_{75}$ |
|---|---|---|---|---|---|---|---|---|
| CBNet-X152 (Liu et al., 2020) | 238M | 1358G | 50.7 | 69.8 | 55.5 | 43.3 | 66.9 | 46.8 |
| Swin-T (Liu et al., 2021) | 86M | 745G | 50.5 | 69.3 | 54.9 | 43.7 | 66.6 | 47.1 |
| RovCol-T (Cai et al., 2023) | 88M | 741G | 50.6 | 68.9 | 54.9 | 43.8 | 66.7 | 47.4 |
| Swin-S (Liu et al., 2021) | 107M | 838G | 51.8 | 70.4 | 56.3 | 44.7 | 67.9 | 48.5 |
| RovCol-S (Cai et al., 2023) | 118M | 833G | 52.6 | 71.1 | 56.8 | 45.5 | 68.8 | 49.0 |
| Swin-B (Liu et al., 2021) | 145M | 982G | 51.9 | 70.5 | 56.4 | 45.0 | 68.1 | 48.9 |
| RovCol-B (Cai et al., 2023) | 196M | 988G | 53.0 | 71.4 | 57.3 | 45.9 | 69.1 | 50.1 |
| ConvNeXt-T | 86M | 741G | 50.8 | 69.4 | 55.2 | 44.5 | 66.9 | 48.5 |
| ConvNeXt-S | 108M | 827G | 51.9 | 71.0 | 56.6 | 45.4 | 68.6 | 49.5 |
| ConvNeXt-B | 146M | 964G | 52.7 | 71.3 | 57.2 | 45.6 | 68.9 | 49.5 |
| CEDNet-NeXt-T | 87M | 753G | 52.5 | 71.4 | 56.8 | 45.9 | 69.0 | 49.7 |
| CEDNet-NeXt-S | 110M | 833G | 53.5 | 72.4 | 58.1 | 46.7 | 69.9 | 50.6 |
| CEDNet-NeXt-B | 148M | 968G | 53.6 | 72.6 | 57.8 | 46.9 | 70.2 | 51.0 |

## 4.3 INSTANCE SEGMENTATION ON COCO

**Settings.** We conducted experiments on instance segmentation with COCO 2017 (Lin et al., 2014) based on the commonly used Mask R-CNN (He et al., 2017) and Cascade Mask R-CNN (Cai & Vas-

Table 3: Results of semantic segmentation on the ADE20K *validation* set. The superscripts $^{ss}$ and $^{ms}$ denote single-scale and multi-scale testing. FPN was adopted as the default fusion module for methods except CEDNet. No extra fusion modules were required after the CEDNet backbone.

| Method | Param. | FLOPs | Input size | mIoU$^{ss}$ | mIoU$^{ms}$ |
|---|---|---|---|---|---|
| Focal-T (Yang et al., 2021) | 62M | 998G | $512^2$ | 45.5 | 47.0 |
| RovCol-T (Cai et al., 2023) | 60M | 937G | $512^2$ | 47.4 | 47.6 |
| Swin-S (Liu et al., 2021) | 81M | 1038G | $512^2$ | 47.6 | 49.5 |
| Focal-S (Yang et al., 2021) | 85M | 1130G | $512^2$ | 48.0 | 50.0 |
| RovCol-S (Cai et al., 2023) | 90M | 1031G | $512^2$ | 47.9 | 49.0 |
| Swin-B (Liu et al., 2021) | 121M | 1188G | $512^2$ | 48.1 | 49.7 |
| Focal-B (Yang et al., 2021) | 126M | 1354G | $512^2$ | 49.0 | 50.5 |
| RovCol-B (Cai et al., 2023) | 122M | 1169G | $512^2$ | 49.0 | 50.1 |
| ConvNeXt-T (Liu et al., 2022) | 60M | 939G | $512^2$ | 46.0 | 46.7 |
| ConvNeXt-S (Liu et al., 2022) | 82M | 1027G | $512^2$ | 48.7 | 49.6 |
| ConvNeXt-B (Liu et al., 2022) | 122M | 1170G | $512^2$ | 49.1 | 49.9 |
| CEDNet-NeXt-T | 61M | 962G | $512^2$ | 48.3 | 48.9 |
| CEDNet-NeXt-S | 83M | 1045G | $512^2$ | 49.8 | 50.4 |
| CEDNet-NeXt-B | 123M | 1184G | $512^2$ | 49.9 | 51.0 |

concelos, 2018) following (Liu et al., 2022; 2021). These two frameworks perform object detection and instance segmentation in a multi-task manner. All training settings were same as Section 4.1.

**Main results.** Table 2 presents the instance segmentation results (see the columns for metrics $AP^m$, $AP_{50}^m$, and $AP_{75}^m$). Based on Mask R-CNN, the models CEDNet-NeXt-T and CEDNet-NeXt-S outperformed their counterparts ConvNeXt-T and ConvNeXt-S by 1.8% and 1.2% mask AP, respectively. When applied to the more powerful Cascade Mask R-CNN, the proposed CEDNet models still yielded 1.3-1.4% mask AP gains over the baseline models. These improvements were consistent with those in object detection. When scaled up to the larger model CEDNet-NeXt-B, CEDNet achieved 46.9% mask AP based on the Cascade Mask R-CNN.

### 4.4 SEMANTIC SEGMENTATION ON ADE20K

**Settings.** We conducted experiments on semantic segmentation with the ADE20k (Zhou et al., 2017) dataset based on UperNet (Xiao et al., 2018) using the MMSegmentation (Contributors, 2020) toolboxes, and report the results on the validation set. The training settings strictly follow (Liu et al., 2022). As the data augmentation strategies used for semantic segmentation were strong enough to train the proposed CEDNet models, no extra data augmentation was introduced.

**Main results.** Table 3 presents the semantic segmentation results. Compared with the ConvNeXt models, CEDNet achieved 0.8-2.2% mIoU gains in the multi-scale test setting with different model variants, which demonstrates the effectiveness of our method in semantic segmentation.

### 4.5 ABLATION STUDIES

To better understand CEDNet, we ablated some key components and evaluated the performance in object detection based on CEDNet-NeXt-T and RetinaNet. Models in Tables 4 and 6 were pre-trained on ImageNet for 100 epochs and fine-tuned on COCO for 12 epochs. The other models were trained under the same settings as Section 4.1.

**Effectiveness of early feature fusion.** To explore the effectiveness of early feature fusion, we constructed several two-stage CEDNet-NeXt-T models varying in fusion time. We modulated the fusion time of each model by adjusting the computational resources allocated to each stage. All models have a similar size. Table 4 shows that the detection accuracy ($AP^b$) gradually improved as the time for multi-scale feature fusion becomes earlier, which demonstrates that early feature fusion is beneficial for dense prediction tasks. Although the two-stage CEDNet models which allocate a proper proportion of computational resources to each stage performed well, we adopted the same

Table 4: Early feature fusion. $n_1^i, n_2^i, n_3^i$ are the number of blocks in the $i$-th stage.

| Time | #Stage | $n_1^1, n_2^1, n_3^1$ | $n_1^2, n_2^2, n_3^2$ | Param | $AP^b$ |
|------|--------|---------|---------|-------|------|
| 6/6 | 2 | 6, 9, 3 | - | 38M | 40.6 |
| 5/6 | 2 | 5, 10, 5 | 1, 2, 1 | 38M | 42.2 |
| 4/6 | 2 | 4, 8, 4 | 2, 4, 2 | 38M | 42.4 |
| 3/6 | 2 | 3, 6, 3 | 3, 6, 3 | 38M | 42.9 |
| 2/6 | 2 | 2, 4, 2 | 4, 8, 4 | 38M | **43.3** |
| 1/6 | 2 | 1, 2, 1 | 5, 10, 5 | 38M | **43.3** |
| 2/6 | 3 | 2, 4, 2 | 2, 4, 2 | 39M | **43.3** |

Table 5: Different token mixers. WA and DW are short for window attention and depth-wise.

| Backbone | Token mixer | Param | $AP^b$ |
|----------|-------------|-------|------|
| ResNet-50 | Vanilla conv $3\times3$ | 38M | 41.7 |
| CEDNet-R50-T | | 39M | **45.3** |
| Swin-T | Local WA | 39M | 44.9 |
| CEDNet-Swin-T | | 37M | **47.4** |
| ConvNeXt-T | DW conv $7\times7$ | 39M | 45.4 |
| CEDNet-NeXt-T | | 39M | **48.3** |
| CSwin-T | Cross WA | 32M | 48.0 |
| CEDNet-CSwin-T | | 33M | **49.5** |

Table 6: Number of stages.

| $m$ | $n_1, n_2, n_3$ | Param | $AP^b$ |
|-----|-----------|-------|------|
| 1 | 6, 9, 3 | 38M | 40.6 |
| 2 | 3, 6, 3 | 38M | 42.9 |
| 3 | 2, 4, 2 | 39M | **43.3** |
| 4 | 1, 4, 1 | 39M | 43.1 |

Table 7: Data augmentation.

| Backbone | Aug. | $AP^b$ |
|----------|------|------|
| ConvNeXt-T | Existing | 45.2 |
| ConvNeXt-T | Ours | 45.4 |
| CEDNet-NeXt-T | Existing | 47.0 |
| CEDNet-NeXt-T | Ours | **48.3** |

Table 8: LR CED block.

| LR block | Param | $AP^b$ |
|----------|-------|------|
| | 38.5M | 47.9 |
| ✓ | 38.6M | **48.3** |

configuration for all stages by default to simplify the structure design and employed three stages to achieve early feature fusion instead (the last row in Table 4).

**Effectiveness on different token mixers.** We constructed the CEDNet models with various token mixers and compared the resulting models with their counterparts (Table 5). The CEDNet models consistently surpassed their counterparts by 1.5-3.6% mAP, which underscores the generality of our CEDNet. *While CEDNet yielded better performance when utilizing the more powerful cross-window attention introduced in CSwin Transformer (Dong et al., 2022), we opted for the more representative ConvNeXt as our baseline in this work and took the $7\times7$ depth-wise convolution from ConvNeXt as the default token mixer for CEDNet.*

**Different numbers of stages.** We built the CEDNet-NeXt-T models with different numbers of stages while maintaining the same configurations across stages. A model with more stages performs multi-scale feature fusion earlier. Table 6 shows that the CEDNet-NeXt-T model with three stages achieved the best detection accuracy. Intuitively, a model with more stages can fuse features more sufficiently, but more network connections may make it harder to optimize.

**Influence of data augmentation.** We compared the data augmentation strategy used by (Liu et al., 2022) and the enhanced data augmentation strategy we adopted for detection fine-tuning. Table 7 shows that the ConvNeXt models achieved similar results under both settings, but our CEDNet model exhibited notable improvements with the enhanced data augmentation. This may be because that CEDNet has a higher capacity than ConvNeXt, and the data augmentation strategy for training the ConvNeXt models are not sufficient to harness the full potential of the CEDNet models.

**Effectiveness of the LR CED block.** Table 8 shows the results of ablation experiments about the LR CED block. The model with LR CED block achieved 0.4% box AP gains over the model without LR CED block. Since the LR CED block only incorporates a lightweight dilated *depth-wise* convolution beyond the standard CED block, it introduces negligible increase in parameters (less than 1%).

## 5 CONCLUSION

We present a universal network named CEDNet for dense prediction tasks. Unlike the widely-used FPN and its variants that usually employ a lightweight fusion module to fuse multi-scale features from pre-designed classification networks, CEDNet introduces several cascade stages to learn multi-scale features. By integrating multi-scale features in the early stages, CEDNet achieves more effective feature fusion. We conducted extensive experiments on several popular dense prediction tasks. The excellent performance demonstrates the effectiveness of our method.

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
