# A    IMAGE CLASSIFICATION ON IMAGENET-1K

Table A1: Results of image classification on the ImageNet-1K *val*. All models were trained and evaluated on 224×224 resolution with the same settings.

| Method | Param. | FLOPs | Top-1 Acc. |
|---|---|---|---|
| HRNet-w32 (Wang et al., 2019) | 38M | 7.6G | 78.4 |
| Swin-T (Liu et al., 2021) | 28M | 4.4G | 81.2 |
| MSG-T (Fang et al., 2022) | 25M | 3.8G | 82.4 |
| FocalNet-T (Yang et al., 2022) | 28M | 4.5G | 82.1 |
| RovCol-T (Cai et al., 2023) | 30M | 4.5G | 82.2 |
| Swin-S (Liu et al., 2021) | 50M | 8.7G | 83.1 |
| MSG-S (Fang et al., 2022) | 56M | 8.4G | 83.4 |
| FocalNet-S (Yang et al., 2022) | 50M | 8.6G | 83.4 |
| RovCol-S (Cai et al., 2023) | 60M | 9.0G | 83.5 |
| Swin-B (Liu et al., 2021) | 88M | 15.4G | 83.4 |
| MSG-B (Fang et al., 2022) | 84M | 14.2G | 84.0 |
| FocalNet-B (Yang et al., 2022) | 89M | 15.4G | 83.9 |
| RovCol-B (Cai et al., 2023) | 138M | 16.6G | 84.1 |
| ConvNeXt-T (Liu et al., 2022) | 29M | 4.5G | 82.1 |
| ConvNeXt-S (Liu et al., 2022) | 50M | 8.7G | 83.1 |
| ConvNeXt-B (Liu et al., 2022) | 89M | 15.4G | 83.8 |
| CEDNet-NeXt-T (Hourglass-style) | 34M | 5.7G | 82.6 |
| CEDNet-NeXt-T (UNet-style) | 34M | 5.7G | 82.9 |
| CEDNet-NeXt-T (FPN-style) | 34M | 5.7G | 83.1 |
| CEDNet-NeXt-S (FPN-style) | 55M | 9.8G | 83.9 |
| CEDNet-NeXt-B (FPN-style) | 95M | 16.2G | 84.3 |

Table A1 shows the results of the CEDNet models in comparison with other methods. The CEDNet models achieved better results than their counterparts, *i.e.*, the ConvNeXt models. This improvement is likely because that the pre-trained CEDNet models have slightly more parameters than their counterparts. The increased number of parameters arises from the fact that these pre-trained CEDNet models are specifically designed for downstream dense prediction tasks. In these tasks, models using CEDNet as their backbone do not need additional fusion modules, which are indispensable for their counterparts. Therefore, we allocated a few extra parameters to the CEDNet models in Table A1 to ensure that all models in subsequent dense prediction tasks have comparable size for a fair comparison (refer to the parameters and FLOPs presented in Table 1 of the paper). *Please note that the CEDNet is specifically designed for dense prediction tasks, and surpassing state-of-the-art methods in image classification is not our goal.*

# B    FURTHER ANALYSIS

We analyzed why CEDNet performed better than its baselines by comparing their input gradient distributions. Specifically, we focused on the images in the COCO *val 2017* where the CEDNet models achieved the most improvements over their counterparts. We fed the selected images into the trained detectors and performed the backward process to acquire the input gradient maps of total detection loss. For each image, we define the *important region* as the region where the absolute gradient is greater than $t$, and $t$ is a threshold to adjust the size of the important region.

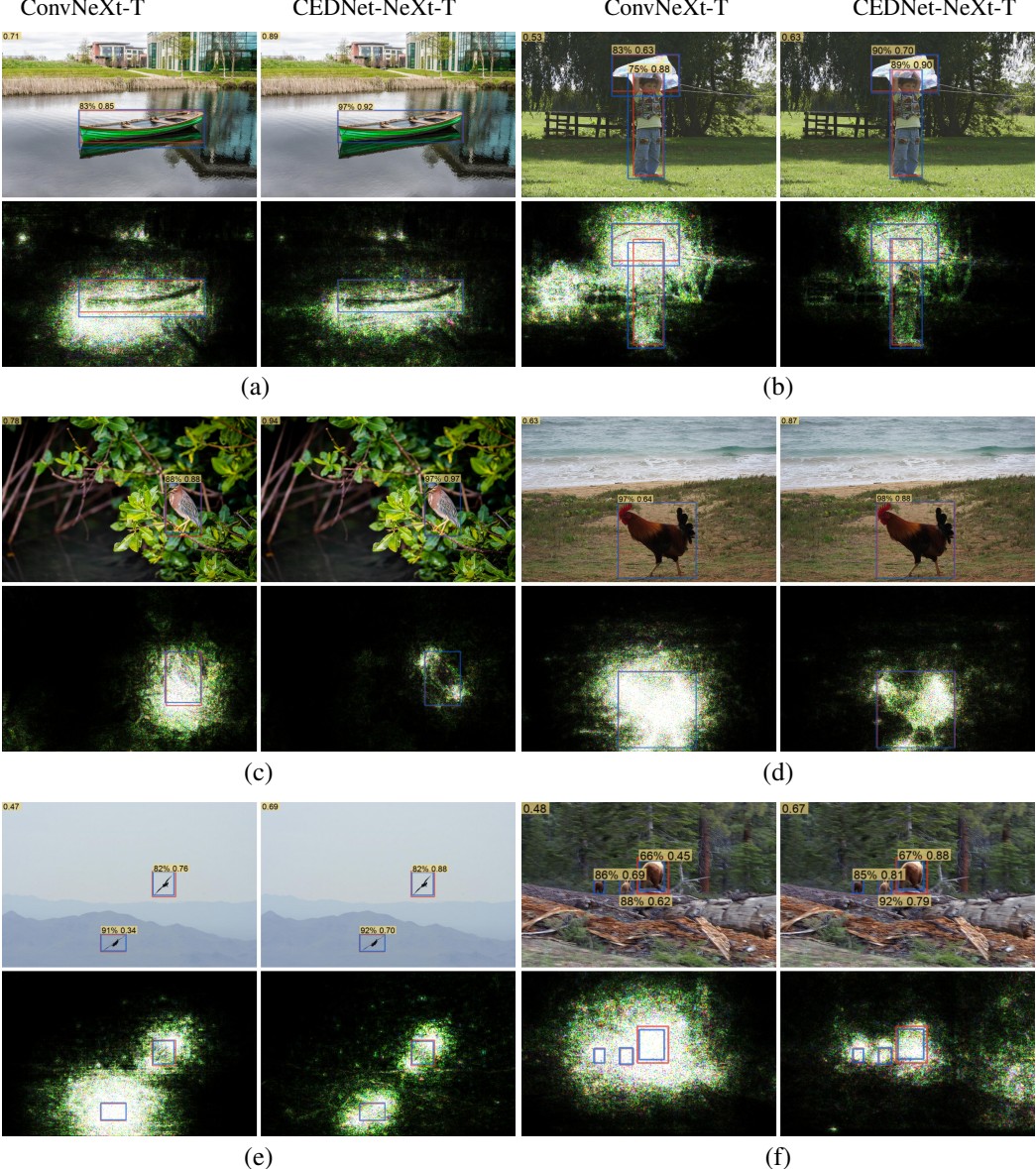

Figure A1: Detection results (top row) and input gradient maps (bottom row) of RetinaNet based on ConvNeXt-T (left column) and CEDNet-NeXt-T (right column). Red boxes represent human-annotated boxes, and blue ones indicate predicted boxes. Above each predicted box, the score pair (iou, confidence) is provided. The image detection quality score is displayed in the top left corner of each image. In the gradient maps, brighter colors signify higher gradients. Please zoom in for a clearer view.

**How to acquire the target images?** For every annotated object in an image, we first acquired the detection result with maximum IoU. If there is no matched result for a specific object, we added a virtually matched result for it and set both the IoU and confidence score of the matched result to zero. We then calculated the *detection quality score* of each annotated object by multiplying the IoU and the confidence score of the matched result. The detection quality score of an image was the average quality score of all objects in the image. Finally, we obtained the top-1000 images where the CEDNet models achieved the most improvements according to the image detection quality scores.

**Results.** We compared CEDNet with ConvNeXt and Swin Transformer based on RetinaNet. Figure A1 shows that the RetinaNet with CEDNet concentrates more on objects with more discriminative boundaries and predicts more precise bounding boxes with higher confidences. In some

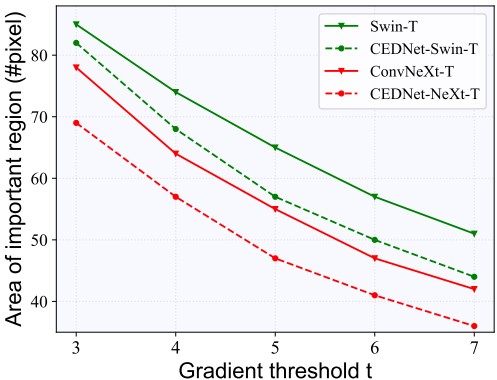

Figure A2: Comparison on the average area of *important regions* generated from RetinaNet models based on various backbones with different gradient thresholds. We normalized the values of both axes for better visualization.

cases (panels (d)-(f)), the IoUs of the boxes predicted by the CEDNet-based RetinaNet are close to that predicted by the ConvNeXt-based RetinaNet, but the confidences of these boxes predicted by the CEDNet-based RetinaNet are higher than that predicted by the ConvNeXt-based RetinaNet. The higher confidences of true positive predictions are more likely to lead to a higher mAP because the confidences of predicted boxes decide the ranking when calculating the precision-recall curve. In addition, we calculated the average area of important regions with different gradient thresholds on those selected images. Figure A2 shows that the average area of important regions generated from the CEDNet-based detectors is smaller than that generated from their counterparts, indicating that the CEDNet models concentrate on smaller regions.