# OpenReview forum: "CEDNet: A Cascade Encoder-Decoder Network for Dense Prediction"
_ICLR.cc/2024/Conference — Submitted to ICLR 2024_

### Official Review · Reviewer_Wmqn · 2023-10-29

**Soundness:** 2 fair
**Presentation:** 2 fair
**Contribution:** 2 fair
**Rating:** 5
**Confidence:** 4

**Summary:**

The paper proposes a new backbone network, CEDNet. It consists of multiple cascade encoder-decoder stages that share the same structure. This allows for early and iterative multi-scale feature fusion.

**Strengths:**

1. A new cascade design allows for early feature fusion.
2. Generalizable to different encoder-decoder styles and token mixers.
3. More streamlined structure compared to other early fusion work.

**Weaknesses:**

1. The novelty of the block design is limited. The CED block and LR-CED block are similar to the block used in ConvNext. It looks like a tiny modification to the existing method. The long-range interactions are not really justified or analyzed.
2. The motivation for the cascade design is not very clearly explained. The authors claim it allows high-level features to guide low-level features, but do not provide analysis into why proposed cascading enables this better than other approaches ($i.e.,$ delayed fusion).

**Questions:**

Please see the weakness part.

**Details Of Ethics Concerns:**

No ethics concern.

---

### Official Review · Reviewer_sm4X · 2023-10-31

**Soundness:** 3 good
**Presentation:** 3 good
**Contribution:** 3 good
**Rating:** 5
**Confidence:** 5

**Summary:**

This paper presents one novel feature fusion mechanism for dense prediction tasks. The previous feature fusion methods, \ie the widely-used FPN and its variants, usually fuse multi-scale features from the pre-trained classification networks. To enhance the effectiveness of the multi-scale feature fusion, this paper designs one end-to-end deep model architecture CEDNet that could integrate the multi-scale features in the early stages.

**Strengths:**

1. Compared to the classical deep methods for dense prediction tasks, the experimental results show that the proposed method CEDNet achieves obvious improvement by integrating the multi-scale features in the early stages.

2. As feature fusion plays one critical role in dense prediction tasks, this paper also proposes one interesting concept ``the time for feature fusion``. This concept could provide one perspective for the community to investigate how to design the earlier time for future fusion.

3. This paper is well-written and easy to understand.

**Weaknesses:**

1. In Table 1,  the flops for the proposed methods may have one typo. 255M->255G?

2. The CED block used in the proposed methods is similar to the ConvNeXt block [1]. Please further describe the difference between these two modules and provide a deeper analysis.

3. Table 1 shows the FPN-style encoder-decoder slightly outperformed the models built on the UNet-style and Hourglass-style encoder-decoder on object detection for the baseline ConvNeXt-T.  Please further compare the proposed three encoder-decoder architectures with the larger baselines ConvNeXt-S and ConvNeXt-B.

4. For the pre-training, besides the last decoder, the author claims that the parameters of the proposed encoder-decoder method CEDNet have been propagated by the Imagenet-1K datasets. For the proposed cascaded encoder-decoder architectures, if there exist some learned layers in the decoder part, these layers have also been pre-trained. For the downstream dense tasks, as shown in Table 2 and Table 3, the compared method ConvNeXt-T only adopts the classical FPN  as the default fusion module. Considering ImageNet-1K has more training images than the COCO and ADE20K, it seems the massive pretraining of the decoder could benefit the performance of the final dense prediction tasks.




[1]. A convnet for the 2020s，CVPR2022.

**Questions:**

This paper proposes one novel feature fusion deep method. However, the proposed method is not well compared with the classical framework (ConvNeXt-T + FPN), and the experiments are not strong to support the contribution of the proposed architecture.

---

### Official Review · Reviewer_XJYk · 2023-11-01

**Soundness:** 3 good
**Presentation:** 2 fair
**Contribution:** 1 poor
**Rating:** 3
**Confidence:** 5

**Summary:**

This work presents a cascade encoder-decoder network for dense prediction tasks. It introduces several cascade stages to extract multi-scale features. With typical blocks, CEDNet achieves effective feature fusions. Experiments on several datasets demonstrates the effectiveness of the proposed method.

**Strengths:**

1.This work is easy to follow, since most of the techniques are based on previous works, such as cascade structure, Hourglass,U-Net and FPN,etc.
2.The proposed method shows better results than some previous methods.

**Weaknesses:**

There are some key concerns:

1.Technical contributions

In fact, this work is largely based on previous works, such as cascade structure, Hourglass,U-Net and FPN, etc. The key differences are the cascade structure. However, in my knowledge, the proposed methods have appeared in previous works, such as Stacked U-Net, Cascade U-Net/FPN. There are only small differences in the block design. The macro idea is the same. There are no essential differences in theory. As for the CED block, the authors just use typical residual modules with recent tricky improvement. I agree that we need more powerful multi-scale feature fusion methods for dense prediction task. However, the methods in this work are not a game-changer for this topic. There are too many existing modules and techniques.

2.Insufficeint experiments

First, there are no full comparisions with recent methods. For the object detection task, the authors only compare with out-of-date methods in Tab.1 (Only one method after 2022). Should the authors provide more reuslts with recent works? I suggest the authors add more comparisions with recent methods. As far as I know, there are many methods perform better than the proposed. In addition, I wonder why the CEDNet-NeXt-T (Hourglass-style/U-Net-style/FPN-style) have the same Param. and FLOPs. I think it is impossible. Second, in Tab.4 and 6, the authors keep the parameter number to verify the effects. I think the direct cascade the structure is more reasonable. How about the results? In fact, if the stage one is very powerful, other stages are not needed. The gains may be too small. Third, there are no enough examples to show the visual results of three tasks. In fact, the authors only present the results in tables Please list the visual results. There are no visual examples for the failure cases. Finally, the LR CED has a very big effect for the performance as shown in Tab.8. I think it weakens the contributions of other modules when compared with the results in Tab.2.

3.Unclear details

There are some unclear details. For examples, how to get 1/8 feature maps with two sequential 3×3 convolutional layers (each with a stride of 2)? Should it be 1/4? What is the effect of using differen dilation rate in LR CED?

**Questions:**

Please see the weakness part.

---

### Official Review · Reviewer_fhWB · 2023-11-15

**Soundness:** 2 fair
**Presentation:** 3 good
**Contribution:** 2 fair
**Rating:** 3
**Confidence:** 5

**Summary:**

This paper introduces CEDNet, a new family of neural networks intended exclusively for dense prediction challenges. The core idea of CEDNet is to advance the time for feature fusion,as this can potentially improve the efficacy of multi-scale feature fusion, according to the authors. The effectiveness is suggested by experimental findings on object detection, instance segmentation, and semantic segmentation.

**Strengths:**

- The motivation is clear, i.e., the authors intuitively believes that dense prediction tasks will benefit from early feature fusion.
- The overall CEDNet architecture is clear-cut and easy to understand.
- Quantitative experimental comparisons indicate useful improvements.

**Weaknesses:**

- After attentively reading the paper, a conflicting impression emerged. Even if there are noticeable improvements over benchmarks, there is yet little concrete proof to support the claim that late fusion is less beneficial, which seriouly undermines the credibility of the motivation and the soundness the whole work. If the authors can locate some concrete evidence to pinpoint the issue, such as a feature map visualization (only a hint instead of remanding), it will greatly enhance the credibility. One such visualization is disappointingly only available in the supplemental file and is not included in the main paper.
- It appears that the authors' understanding of low- and high-level features is flawed. Low-level feature is not identical to high-resolution feature and high-level feature is not the low-resolution one. In CV field, corners, edges, textures, and colors are examples of low-level features, whereas items, scenes, and behaviors are examples of high-level features. High-level features typically originate from deeper layers, while low-level features are typically found in earlier layers. In the conventional pyramid structure of modern neural networks, high-level feature happens to be the low-resolution one. Thus, the key claim is questionable that the CEDNet design can incorporate high-level features from early stages to guide low-level feature learning in subsequent stages.
- Stacked encoder-decoder is not something fresh, e.g., CascadePSP [1], SDN [2].

[1] CascadePSP: Toward Class-Agnostic and Very High-Resolution Segmentation via Global and Local Refinement. CVPR 2020.

[2] Stacked Deconvolutional Network for Semantic Segmentation. TIP 2019.

**Questions:**

1. The way CEDNet uses backbone features for dense prediction tasks is unclear. Is each stage's final feature map utilized?
2. It is highly peculiar that the number of parameter and FLOPs are the same for each CEDNet style in Table 1.
3. Why does the FPS not change when the FLOPs of CEDNet are drastically decreased by three orders of magnitude, as Table 1 illustrates? And three orders of magnitude appears improbable.
4. For semantic segmentation, SDN [1] also employs a stacked encoder-decode structure. The authors ought to add in-depth textual analysis and comparative experiments.
5. The concept of EPRNet [2], which distributes multi-scale fetaure fusion in the early backbone stages, is comparable to that of the proposed CEDNet. Textual discussion ought to be complemented.
6. Lack of comparison with the latest methods, e.g., InternImage [3].

It may require a significant rewrite to make this manuscript more sound.

[1] Stacked Deconvolutional Network for Semantic Segmentation. TIP 2019.

[2] EPRNet: Efficient pyramid representation network for real-time street scene segmentation. TITS 2021.

[3] InternImage: Exploring Large-Scale Vision Foundation Models with Deformable Convolutions. CVPR 2023.

---

### Meta-Review · Area_Chair_vvFM · 2023-12-09

**Metareview:**

The meta-reviewer has carefully read the paper, reviews, rebuttals, and discussions between authors and reviewers. The meta-reviewer agrees with the reviewers that this submission is below the bar of ICLR. The work introduces CEDNet, a cascade encoder-decoder network for dense prediction tasks that utilize multi-scale features and achieve effective feature fusions, showing promising results on several datasets. However, as pointed out by the reviewers, the paper primarily builds upon existing techniques like cascade structures and residual modules without significant innovation. The experiments are also seen as insufficient, lacking comparisons with up-to-date methods and adequate visual results. Moreover, the impact of the learning rate on CEDNet's performance raises questions about the actual effectiveness of the proposed modules. Details about the network's implementation are also unclear, casting doubt on the reported improvements. The meta-reviewer cannot recommend acceptance at the moment.

**Justification For Why Not Higher Score:**

N/A

**Justification For Why Not Lower Score:**

N/A

---

### Decision · Program_Chairs · 2024-01-16

Reject